# Learning Diverse and Physically Feasible Dexterous Grasps with Generative Model and Bilevel Optimization

**Albert Wu**
Computer Science Department
Stanford University, United States
`amhwu@stanford.edu`

**Michelle Guo**
Computer Science Department
Stanford University, United States
`mguo95@cs.stanford.edu`

**C. Karen Liu**
Computer Science Department
Stanford University, United States
`karenliu@cs.stanford.edu`

**Abstract:** To fully utilize the versatility of a multi-fingered dexterous robotic hand for executing diverse object grasps, one must consider the rich physical constraints introduced by hand-object interaction and object geometry. We propose an integrative approach of combining a generative model and a bilevel optimization (BO) to plan diverse grasp configurations on novel objects. First, a conditional variational autoencoder trained on merely six YCB objects predicts the finger placement directly from the object point cloud. The prediction is then used to seed a nonconvex BO that solves for a grasp configuration under collision, reachability, wrench closure, and friction constraints. Our method achieved an $86.7\%$ success over 120 real world grasping trials on 20 household objects, including unseen and challenging geometries. Through quantitative empirical evaluations, we confirm that grasp configurations produced by our pipeline are indeed guaranteed to satisfy kinematic and dynamic constraints. A video summary of our results is available at `youtu.be/9DTrImbN99I`.

**Keywords:** Dexterous grasping, Grasp planning, Bilevel optimization, Generative model

## 1 Introduction

Performing diverse grasps on a variety of objects is a fundamental skill in robotic manipulation. Diverse grasp configurations allow flexible interaction with the objects while satisfying requirements demanded by the higher level task of interest. *Dexterous grasping*, which refers to object grasping with a fully actuated, multi-finger dexterous robotic hand, has the potential to achieve diverse grasp configurations with applicability to a wide range of objects. This is in contrast with *simple grasps* achieved by parallel jaw grippers or underactuated multi-finger grippers, both of which have fingertip workspaces restricted to a subspace of the 3D task space.

We identify two major challenges in planning diverse dexterous grasps. Firstly, the solution space is multimodal due to the many finger placement possibilities and the lack of a metric defining an optimal grasp among valid answers. Secondly, each valid grasp is governed by nonconvex physical constraints such as collision, contact, and force closure.

The multimodality of dexterous grasp planning motivates the use of deep learning-based approaches. It is challenging to produce diverse and multimodal grasp plans with regression or naive supervised methods. Analytical approaches such as precomputing a grasp library or sampling-based grasp planning, while popular for planning simple grasps, are computationally intractable on high-dimensional dexterous grasps. Nevertheless, learning to plan grasps that strictly obey physical constraints is difficult. Model-based optimization has historically been applied to enforce exact constraints. However,

6th Conference on Robot Learning (CoRL 2022), Auckland, New Zealand.

in practice the dexterous grasp planning problem is arguably too nonconvex to solve directly. Consequently, previous works usually relax the constraints as scalar losses and solve the relaxed problem, sacrificing exact constraint satisfaction guarantees for practicality.

We propose an integrative approach that combines learning and optimization to produce diverse physically-feasible dexterous grasp configurations for unseen objects. Our method first predicts an initial set of contact points using a conditional variational autoencoder (CVAE). The contact points are then projected onto the manifold of kinematically and dynamically feasible grasps by solving a bilevel optimization (BO) problem. Our key contributions are summarized below:

• **Learning-based dexterous grasp planning pipeline** that integrates CVAE and BO to produce diverse, fully-specified, and constraint-satisfying dexterous grasps from point clouds.

• **Bilevel grasp optimization formulation** that takes a learning-based grasp prediction and outputs a dexterous grasp that satisfies reachability, collision, wrench closure, and friction cone constraints.

• **Successful and extensive hardware validation** on 20 household objects. Our method achieved an overall success rate of 86.7% over 120 grasp trials.

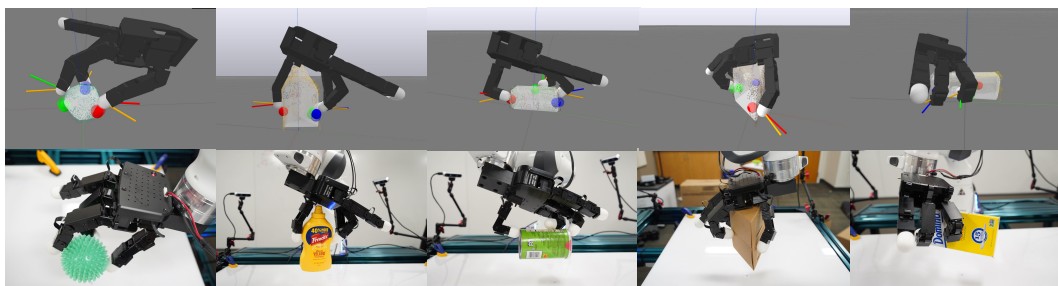

Figure 1: Our method can pick up different objects shapes with a diverse set of grasp configurations.

## 2 Related Work

We limit our discussion to literature on grasping in uncluttered scenes, with an emphasis on dexterous grasping. We exclude work that focuses on object segmentation in cluttered environments or other types of manipulation, such as in-hand manipulation.

### 2.1 Dexterous Grasping

Dexterous grasping is a long standing problem in the robotics community. In general, the literature can be split into learning-based and analytical methods. Early learning-based grasp planners seek to fit the space of possible grasps for rapid grasp generation (e.g., [1]). More recent papers have shifted to producing grasps directly with complex model architectures, such as generative models [2, 3, 4] and convolutional neural networks [5, 6]. However, most of these works do not account for physical laws and have restricted, or even missing, hardware evaluation. This issue is exacerbated by the difficulty of simulating contact-rich interactions [7, 8]. The quality of grasps produced by learning exclusively with simulation is questionable. Moreover, many papers do not emphasize the ability to learn *diverse* grasps, thus forgoing a key benefit of dexterous hands over simple grippers. Notably, some publications on learning dexterous grasping originate from computer vision and graphics communities (e.g., [9, 10, 11, 12, 13, 14]). While these papers cover diverse grasp generation on fully dexterous hands, their ultimate objective tends to be achieving photorealistic human grasps in simulation rather than satisfying strict real-world physical constraints.

On the other hand, analytical dexterous grasp planners are often inspired by physical constraints such as force closure, friction cone, robustness to external disturbance wrenches, and object contact (e.g., [15, 16, 17, 18]). Historically, analytical methods are standalone and includes both grasp modality exploration and physical constraint compliance. We point the readers to [19, 20] for a detailed review on these methods. Some limitations of these approaches include dependency on a high-fidelity object model and lack of diversity in the generated grasps [20].

In recent years, some learning-based approaches have adopted a "grasp refinement" step motivated by analytical metrics [4, 18, 21]. Nevertheless, these metrics tend to be formulated as "relaxed" op-

timizations to keep the problem tractable. Instead of enforcing the constraints directly, the violation of each constraint is cast as a scalar loss and summed together. This results in an unconstrained optimization which is significantly easier to solve. However, there is no guarantee that the optimization output will indeed satisfy the constraints that motivated the loss design.

## 2.2 Simple Grasping

Due to the geometry of parallel jaw grippers, simple grasp planning can be reduced to computing a 6-DoF gripper pose in space. As such, analytical simple grasp planners may directly reason about the object geometry (e.g., [22]) or rank grasps using grasp quality metrics [19]. Recently, learning-based approaches that predict wrist poses have gained significant traction (e.g., [23, 24, 25]). We refer the readers to [26, 19, 20, 27, 28] for a thorough review. These approaches seldom scale directly to dexterous grasp planning, which is a significantly more complex problem.

## 2.3 Bilevel Optimization (BO) for Planning

While BO theory is established in literature [29], application of BO on motion planning is relatively new. BO has been applied to continuous systems [30], robotic locomotion [31, 32], simple pushing and pivoting [33], collaborative object manipulation [34], and task and motion planning [35]. To our best knowledge, this work is the first application of BO on dexterous robotic manipulation.

## 3 Method

Our method consists of a learned model that predicts a plausible finger placement $\mathcal{P} \in \mathbb{R}^{3 \times 3}$, and an analytical process that computes a *physically feasible dexterous grasp* $q^* \in \mathbb{R}^{22}$ (Definition 1) guided by $\mathcal{P}$. At inference time, the pipeline takes in an object point cloud and outputs a fully specified grasp configuration $q^*$. Figure 2 gives an overview of our method. We assume the object is grasped with exactly 3 fingers. This section first formally specify the modeled physical constraints, then discuss each of the components in the pipeline. Our implementation is available at `github.com/Stanford-TML/dex_grasp`.

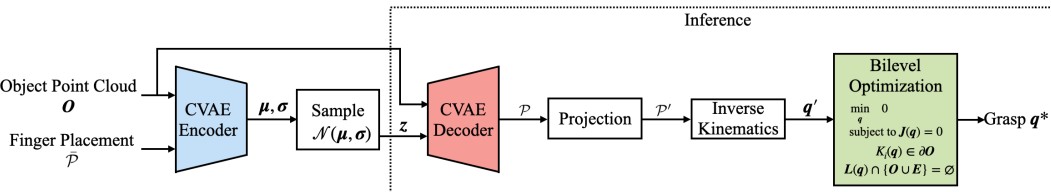

Figure 2: Overview of our method. We train a CVAE that predicts finger placements $\mathcal{P} \in \mathbb{R}^{3 \times 3}$ given an object point cloud $O$. At inference time, we first obtain a finger placement prediction $\mathcal{P}$, which is not guaranteed to be physically feasible. Next, we compute a grasp configuration initial guess $q' \in \mathbb{R}^{22}$ from $\mathcal{P}$. Finally, we apply BO to compute a physically feasible grasp $q^* \in \mathbb{R}^{22}$.

**Definition 1** (Physically Feasible Dexterous Grasp). Given an object $O$ in environment $E$, a grasp configuration $q$, and a desired finger placement $\mathcal{P} = (p_1, p_2, p_3) \in \mathbb{R}^{3 \times 3}$, we consider $(O, E, q, \mathcal{P})$ to be *physically feasible* if they satisfy following dynamic and kinematic constraints. In our setup, $q \in \mathbb{R}^{22}$ and $E$ is the fixed tabletop (see Section 4.1).

## 3.1 Dynamic Constraints: Wrench Closure and Friction Cone

For a static dexterous grasp, the dynamic constraints include the wrench closure constraint and the friction cone constraint. The wrench closure constraint requires the sum of the external wrench from all contact points to be zero: $\sum_{i=1}^{3} f_i = 0$ and $\sum_{i=1}^{3} p_i \times f_i = 0$. $f_i \in \mathbb{R}^3$ is the unknown contact force applied at position $p_i \in \mathbb{R}^3$ from the dexterous hand to the object.

Given the static friction coefficient $\mu$ and the outward-pointing surface normal $\hat{n}_i$, the friction cone constraint requires the normal force to be nonnegative and the contact force to be within the friction cone. Using a polyhedral cone approximation ([36]) with an orthogonal basis $\hat{t}_{i,j} \perp \hat{n}_i, \forall j \in \{1, 2\}$,

we arrive at the following approximated friction cone constraints:

$$0 \leq -\boldsymbol{f}_i \cdot \hat{\boldsymbol{n}}_i \quad \text{and} \quad |\boldsymbol{f}_i \cdot \hat{\boldsymbol{t}}_{i,j}| \leq -\mu \boldsymbol{f}_i \cdot \hat{\boldsymbol{n}}_i, \forall i \in \{1,2,3\}, \forall j \in \{1,2\}. \tag{1}$$

A grasp is *dynamically feasible* if it satisfies both wrench closure and friction cone constraints. Leveraging the polyhedral approximation, dynamic feasibility can be cast as a quadratic program (QP):

$$\min_{\boldsymbol{f}_1, \boldsymbol{f}_2, \boldsymbol{f}_3} \|\sum_{i=1}^{3} \boldsymbol{f}_i\|_2^2 + \|\sum_{i=1}^{3} \boldsymbol{p}_i \times \boldsymbol{f}_i\|_2^2 \quad \text{subject to } 0 < f_{min} \leq -\boldsymbol{f}_i \cdot \hat{n}_i \text{ and (1)}. \tag{2}$$

The optimization in Equation (2) has objective value 0 if and only if the grasp is dynamically feasible. Note that we set an arbitrary lower bound $f_{min} \in \mathbb{R}^+$ on the normal force to avoid the trivial solution of $\boldsymbol{f}_i = \boldsymbol{0}$. The exact value of $f_{min}$ is irrelevant because changing $f_{min}$ represents scaling the optimal contact forces, which does not affect membership in the friction cone.

## 3.2 Kinematic Constraints: Reachability and Collision

The kinematic constraints include reachability constraints and collision constraints. Reachability constraints enforce the kinematic ability of the robot hand to reach the contact point $\boldsymbol{p}_i$ on the object surface $\partial \boldsymbol{O}$, i.e. $\exists \boldsymbol{q} : K_i(\boldsymbol{q}) = \boldsymbol{p}_i \in \partial \boldsymbol{O}, \ \forall i \in \{1,2,3\}$. Here $K_i : \mathbb{R}^{22} \mapsto \mathbb{R}^3$ is the forward kinematics function that maps the hand state $\boldsymbol{q}$ to the position of the $i$-th fingertip. Assuming only fingertip contacts, collision constraints ensure that no robot link $\boldsymbol{L}(\boldsymbol{q})$ is in collision with the object $\boldsymbol{O}$ except at the fingertip or with the environment $\boldsymbol{E}$, i.e. $\boldsymbol{L}(\boldsymbol{q}) \cap \{\boldsymbol{O} \cup \boldsymbol{E}\} = \emptyset$. Both reachability and collision constraints are nonconvex constraints.

## 3.3 Learning to Predict Finger Placement from a Physically Feasible Grasp Dataset

We leverage a generative model to sample potential finger placements $\mathcal{P}$ given an arbitrary object observed as a point cloud $\hat{\boldsymbol{O}}$. An immediate challenge faced by the learning approach is the lack of large-scale datasets for dexterous robotic hands. While datasets for real human hands do exist, retargeting human grasping configurations to a robot hand with different kinematic structures and joint limits presents numerous challenges. We opt to synthesize a large-scale dataset of physically feasible grasping configurations for the Allegro robot hand used in this paper.

### 3.3.1 Creating a Physically Feasible Dexterous Grasp Dataset

We generate physically feasible grasps of six YCB objects [37] in simulation. The chosen objects are "cracker box," "sugar box," "tomato soup can," "mustard bottle," "gelatin box," and "potted meat can." We consider a realistic scenario where the object is placed on a flat surface instead of free floating. As such, we need to consider different object rest poses of on the surface in addition to finger placements. In summary, we first generate random object rest poses and enumerate all possible finger placements $\bar{\mathcal{P}}$ using the corresponding simulated object point cloud $\hat{\boldsymbol{O}}$. If a kinematically and dynamically feasible grasp $\boldsymbol{q}$ can be found for $\bar{\mathcal{P}}$, $(\hat{\boldsymbol{O}}, \boldsymbol{q}, \bar{\mathcal{P}})$ is added to the dataset $\mathbb{P}$. The detailed procedure is described in the supplementary material.

### 3.3.2 Training a Conditional Variational Autoencoder (CVAE)

To compute diverse grasps for arbitrary objects directly from point cloud observations, we predict fingertip contact points on the object surface with a conditional variational autoencoder (CVAE) [38]. Our model is adapted from [24] and consists of an encoder $E$ and a decoder $D$ that are based on the PointNet++ architecture [39]. The model seeks to maximize the likelihood of producing a set of contact points $\mathcal{P}$ deemed feasible in Section 3.3.1 given the point cloud $\hat{\boldsymbol{O}}$.

The encoder $E(\boldsymbol{z} \mid \bar{\mathcal{P}}, \hat{\boldsymbol{O}})$ maps a grasp $\bar{\mathcal{P}}$ and a point cloud $\hat{\boldsymbol{O}}$ to the latent space. We assume the latent variable has a normal distribution $\mathcal{N}(\boldsymbol{0}, \mathcal{I})$. Meanwhile, given a latent sample $\boldsymbol{z} \sim E$, the decoder attempts to reconstruct the finger placement $\mathcal{P}$. We seek to minimize the element-wise $L^1$-norm reconstruction loss $L_{rec}(\mathcal{P}, \bar{\mathcal{P}}) \triangleq \|\mathcal{P} - \bar{\mathcal{P}}\|_1$ for a feasible grasp from the dataset $\bar{\mathcal{P}} \in \mathbb{P}$. Additionally, a KL divergence loss $\mathcal{D}_{KL}$ is applied on the latent distribution $E(\cdot \mid \cdot)$ to ensure a normally distributed latent variable. The complete loss function of the network is defined as

$$L \triangleq \sum_{\boldsymbol{z} \sim E, \bar{\mathcal{P}} \in \mathbb{P}} L_{rec}(\mathcal{P}, \bar{\mathcal{P}}) + \alpha \mathcal{D}_{KL}\left(E(\boldsymbol{z} \mid \bar{\mathcal{P}}, \hat{\boldsymbol{O}}) \, \big\| \, \mathcal{N}(\boldsymbol{0}, \mathcal{I})\right). \tag{3}$$

At inference time, $E$ is removed and a latent sample $z$ is drawn from $\mathcal{N}(\mathbf{0}, \mathcal{I})$. This is passed into the decoder $D(\mathcal{P}|\hat{O}, z)$ along with the point cloud $\hat{O}$ to produce the grasp point prediction $\mathcal{P}$.

## 3.4 Computing Grasps with Physical Feasibility Guarantees using Bilevel Optimization (BO)

While the grasps in the training dataset are physically feasible by construction, there is no guarantee that the CVAE output, $\mathcal{P}$, is physically feasible. Additionally, $\mathcal{P}$ only specifies the finger placement instead of the full hand configuration. We propose a BO to obtain a physically feasible grasping pose $q$ given $\mathcal{P}$. To seed the BO with $\mathcal{P}$, we first obtain an Euclidean projection of $\mathcal{P}$ onto $\partial O$, denoted as $\mathcal{P}' \triangleq \left\{ p_i' \mid p_i' = \arg\min_{p_o \in \partial O} \|p_o - p_i\|_2, \ i \in \{1, 2, 3\} \right\}$. Next, we solve an inverse kinematics (IK) problem for a grasp configuration finger placement problem specified by $\mathcal{P}'$ up to a numerical tolerance $\epsilon$, i.e. find $q' : \|K_i(q') - p_i'\|_2 \leq \epsilon, \ K_i(q') \in \partial O, \ \forall i \in \{1, 2, 3\}$. $K_i : \mathbb{R}^{22} \mapsto \mathbb{R}^3$ is the forward kinematics function of the $i$th finger. $K_i(q') \in \partial O$ is implemented as a constraint on the finger-object signed distance $D(p, O) \in [d_{min}, d_{max}]$. The IK solution $q'$ serves as the initial guess for the BO.

### 3.4.1 Formulating the Bilevel Optimization Grasp Refinement

Accounting for the kinematic and dynamic constraints, the naïve formulation of the grasp optimization problem is given in (4).

$$\min_{q, f_1, f_2, f_3} \quad 0$$

$$\text{subject to} \quad K_i(q) \in \partial O, \ L(q) \cap \{O \cup E\} = \emptyset, \ \|\sum_{i=1}^{3} f_i\|_2 = 0, \ \|\sum_{i=1}^{3} K_i(q) \times f_i\|_2 = 0, \quad (4)$$

$$f_{min} \leq -f_i \cdot \hat{n}_i, \ |f_i \cdot \hat{t}_{i,j}| \leq -\mu f_i \cdot \hat{n}_i, \ \forall i \in \{1, 2, 3\}, \forall j \in \{1, 2\}.$$

Equation (4) does not solve well in practice due to its complexity and nonconvexity. Additionally, while $q'$ may serve as an initial guess for $q$, the initial guess for $f$ is not obvious. Naively relaxing Equation (4), e.g., replacing the objective with $\min_{q, f_1, f_2, f_3} \|\sum_{i=1}^{3} f_i\|_2^2 + \|\sum_{i=1}^{3} K_i(q) \times f_i\|_2^2$, may produce suboptimal $f$ and, consequently, incorrect conclusion on grasp's dynamic feasibility.

We propose leveraging *bilevel optimization* [29, 32] to offload the dynamic feasibility computation from the main optimization. Define $J(q)$ as the minimum objective value of the dynamic constraint QP in Equation (2) with $p_i = K_i(q)$:

$$J(q) \triangleq \min_{f_1, f_2, f_3} \quad \|\sum_{i=1}^{3} f_i\|_2^2 + \|\sum_{i=1}^{3} p_i \times f_i\|_2^2 \quad \text{subject to} \ 0 < f_{min} \leq -f_i \cdot \hat{n}_i \text{ and } (1). \quad (5)$$

We observe that the force closure constraint $J(q) = 0$ can be abstracted away to form a "lower-level" QP. $J(q) = 0$ can be solved to optimality with exiting QP solvers without reliance on a good initial guess for $f_i$.

Applying this abstraction to Equation (4) yields the upper-level problem:

$$\min_{q} \quad 0 \quad \text{subject to} \quad J(q) = 0, \ K_i(q) \in \partial O, \ L(q) \cap \{O \cup E\} = \emptyset. \quad (6)$$

Equation (6) is a bilevel optimization as $J(q) = 0$ is a constraint on the minimizer of another optimization problem (Equation (6)). While Equation (6) still defines a nonconvex optimization, the choice for $f_i$ has been abstracted away entirely to the lower level QP solver. By construction, a valid solution to Equation (6), denoted as $q^*$, defines a physically feasible grasp.

### 3.4.2 Solving the Bilevel Optimization Grasp Refinement

In practice, to solve Equation (6) with a nonconvex optimizer (e.g., SNOPT [40]), one needs to obtain the gradient of each of the constraints with respect to $q$. The gradients of the kinematic constraints can be obtained from simulators with automatic differentiation capabilities (e.g., Drake [41]). The gradient of $J(q)$, $\nabla_q J(q)$, can obtained using differentiable optimization [42]. At a high level, differentiable optimization computes $\nabla_q J(q)$ by taking the matrix differentials of the KKT conditions of the optimization problem at its solution.

We emphasize that the bilevel optimization in Equation (6) is still a highly nonconvex optimization, and the optimizer will likely return a locally optimal $q^*$ in the vicinity of $q'$. Intuitively, this

could be interpreted as optimizing within the grasp family of the CVAE prediction (e.g., "left side grasp" or "top-down grasp"), which implies the necessity of CVAE in the process. This is supported empirically by our ablation studies in Section 4. In practice, the bilevel optimization provides a certification of physical feasibility for the CVAE prediction. A poor $q'$ choice will likely results in the nonconvex optimizer returning infeasibility. This serves as the condition to reject $\mathcal{P}$ and generate a different CVAE grasp prediction with another latent variable sample.

## 4  Experiments

Our trained CVAE achieved a test reconstruction error of $0.5$cm. More training details are available in the supplementary material. The effects of applying BO is shown in Figures 3a and 3b. The finger placement prior to BO cannot form force closure as $\hat{n}_i \cdot \hat{n}_j > 0, \forall i, j \in \{1, 2, 3\}$. BO shifted the thumb and middle fingers to an antipodal configuration, which allows for force closure. This validates our approach's ability to make an initially infeasible grasp prediction physically feasible. The rest of this section focuses on hardware experiments and evaluations.

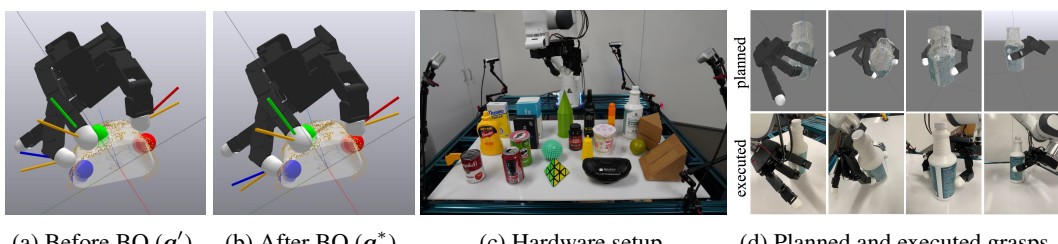

(a) Before BO ($q'$).    (b) After BO ($q^*$).    (c) Hardware setup.    (d) Planned and executed grasps.

Figure 3: **3a and 3b: Before and after BO**, on a bottom-up view of a mustard bottle grasp. Using the color encoding of red: thumb, green: index, and blue: middle, we show $\mathcal{P}$ (solid spheres), $\mathcal{P}'$ (transparent spheres) and $\hat{n}_i$ (colored lines). The direction of the computed contact forces $f_i$ are shown at the respective fingertip with yellow lines. The mismatch between $K(q')$ and $\mathcal{P}'$ is due to the numerical tolerance $\epsilon$. **3c: Hardware setup.** See Section 4.1 for more details. **3d: Grasp sim-to-real.** Correspondence between the planned grasp in simulation and execution on hardware.

### 4.1  Hardware Setup

The hardware setup and evaluated objects are shown in Figure 3c.
**Hardware.** A 16-DoF Allegro v4.0 right hand was used for hardware grasping experiments. The Allegro hand was mounted on a 7-DoF Franka Emika Panda arm to realize the planned wrist pose. The object point cloud was captured by 4 stationary Intel RealSense D435 depth cameras. The implementation details are available in the supplementary material.
**Evaluated Objects.** We evaluated our method on 20 rigid household objects resting on the table. We categorize our objects into three sets:
• **3 Seen objects.** (Leftmost column) mustard bottle, soup can, and sugar box from the training set.
• **4 Familiar objects.** (Second-from-left column) Unseen objects with geometries similar to training objects. Includes boxes (webcam box, mask box) and cylinders (chip can, soda can).
• **13 Novel objects.** Objects whose geometries are distinct from that of any object in the training set. From left to right, front to back: tetrahedron, massage ball, castle, pill bottle, glasses case, condiment bottle, hairspray bottle, ramen, lego, sandwich box (side), pear, sandwich box (upright), alcohol bottle.

### 4.2  Experiment Procedure

At the start of each trial, the object is placed in a specified pose in front of the robot. After observing the point cloud and computing a grasp, the Franka arm first brings the hand to the desired wrist pose using an *ad hoc* trajectory planner. The finger joint angles $\theta^* \in \mathbb{R}^{16}$ from $q^*$ is then executed on the Allegro hand. Contact forces are provided by squeezing the fingertips according to the planned contact forces: $\theta^* \leftarrow \theta^* + k(\nabla_\theta p_i)^T f_i$. Here $\nabla_\theta p_i$ is the Jacobian of the fingertip locations with respect to $\theta$, and $k$ is a fixed "stiffness" constant as motivated by impedance control. The Franka then attempts to lift the object to a fixed height approximately $43$cm above the table. A trial is

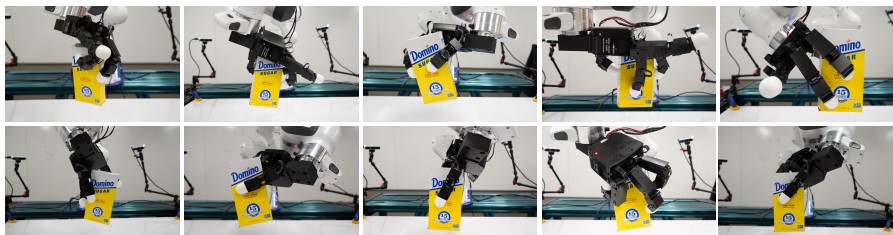

Figure 4: Diverse grasps generated by our method. Each image is a unique grasp generated from different sampled latent variables. All grasps were computed for the same initial object rest pose.

considered successful if the object is lifted and all three fingers remain in contact with the object. 3 repeats are performed for hardcoded policy trials as they are deterministic up to the object point cloud observation. All other trials are repeated 6 times. Our hardware pipeline was implemented with the intent to execute our planned grasp as accurately as possible. Figure 3d illustrates the simulation-hardware grasp correspondence.

## 4.3 Baseline and Ablation Studies

We compare our method against the following approaches. We excluded "BO only" in our ablation studies as it does not return a result without CVAE in practice. BO seldom returns a solution if seeded with a kinematically infeasible solution (e.g., open hand).
• **Hardcoded grasp baseline.** Using the set of seen objects, we designed a top-down tripod grasp policy that chooses the wrist pose based on the point cloud and executes a fixed grasp.
• **CVAE-only as an ablation study.** We solved a collision-free inverse kinematics problem that attempts to match the CVAE predicted fingertip positions $\mathcal{P}$. This mimics a "learning only" approach.
• **CVAE-kinematic as an ablation study.** We ablated the "lower level" part of the optimization (Equation (5)) away and use $q'$ directly without bilevel optimization. This ablation considers kinematic constraints but not dynamic constraints.

## 4.4 Grasp Trial Results

**Our method achieved an overall success rate of 86.7%** over 120 grasp trials on 20 objects. This is superior to the hardcoded baseline, which achieved an overall success rate of $53.3\%$. On seen objects, our CVAE-only ablation achieved a $38.9\%$ success rate and our CVAE-kinematic ablation achieved an $88.9\%$ success rate. The results are summarized in Table 1, and the details are available in the supplementary material. A video summary is available at `youtu.be/9DTrImbN99I`.

**Our method can grasp challenging novel objects.** This includes objects that are difficult for parallel grippers and suction cups to grasp. We discuss the challenges of some of our test objects in the supplementary material. The hardcoded baseline nearly always succeeds on objects that allow top-down grasps and are similar in size to the seen object used for tuning. However, the baseline fails on objects that either require a different grasp type or have significantly different size.

**Our method can generate diverse grasps through sampling different latent $z$'s.** Figure 4 shows 10 distinct successful grasps performed on the upright sugar box.

**The median time to produce a grasp with our method is 14.4 seconds.** The median repeats for each component of our method when generating a grasp is two CVAE $z$ samples, two IK solves for $q'$, and two bilevel optimizations for $q^*$. Detailed timing and repetition results are available in the supplementary material.

Table 1: Object grasping statistics. Our method achieved superior grasp success rate compared to the ablations and the hardcoded baseline.

| Category (object count) | Seen (3) | | Familiar (4) | | Novel (13) | | Overall (20) | |
|---|---|---|---|---|---|---|---|---|
| Ours | 17/18 | 94.4% | 23/24 | 95.8% | 64/78 | 82.1% | 104/120 | 86.7% |
| CVAE only | 7/18 | 38.9% | - | - | - | - | - | - |
| CVAE+kinematics | 16/18 | 88.9% | - | - | - | - | - | - |
| Hardcoded | 8/9 | 88.9% | 9/12 | 75.0% | 15/39 | 38.5% | 32/60 | 53.3% |

Table 2: Evaluation of physical constraint on various grasps. A physically feasible grasp should satisfy $D(\boldsymbol{p}, \boldsymbol{O}) \in [d_{min}, d_{max}] = [-0.68, -0.32]$ and zero force and torque ratios.

| | Median $D$ | Min $D$ | Max $D$ | Max violation | Force ratio | Torque ratio |
|---|---|---|---|---|---|---|
| Ours (all objects) | -0.32 | -0.68 | -0.24 | 0.08 | (0.00, 0.02) | 0.01 (0.00, 2.68) |
| CVAE-kinematics (seen objects) | -0.32 | -0.64 | -0.32 | 0.04 | 0.00 (0.00, 0.12) | 0.01 (0.00, 19.54) |
| CVAE-only (seen objects) | 0.33 | -1.10 | 1.71 | 2.03 | - | - |

Table 3: Correlation between dynamic feasibility and hardware success. Only the grasps plans with wrench closure resulted in successful hardware execution.

| | Successful | | Failed | |
|---|---|---|---|---|
| | Force ratio | Torque ratio | Force ratio | Torque ratio |
| BO rejected (12 of 12 failed) | - | - | 61.81 (25.51, 91.65) | 48.53 (39.89, 78.68) |
| CVAE-kinematics (7 of 12 failed) | 0.00 (0.00, 0.00) | 0.00 (0.00, 0.01) | 50.71 (19.13, 128.85) | 36.75 (30.48, 131.82) |

## 4.5 Quantitative evaluation of physical constraint enforcement

Table 2 summarizes the quantitative evaluation of physical constraints.
**Kinematic constraints.** The maximum finger-object distance constraint violation in a grasp planned by our method is 0.08cm, which is negligible in practice as it is smaller than other error sources such as camera observation error. The CVAE-kinematics ablation, which only considers kinematic constraints, achieved comparable results. The CVAE-only ablation has a maximum violation of 2.032cm, confirming that while CVAE alone can produce qualitatively correct grasp, it cannot enforce kinematic constraints precisely.
**Dynamic constraints**. To evaluate dynamic constraint satisfaction, we chose "force (torque) ratio" as our metric, i.e. $\frac{\|\sum_{i=1,2,3} \boldsymbol{f}_i\|_2}{\frac{1}{3}\sum_{i=1,2,3}\|\boldsymbol{f}_i\|_2} \times 100\%$, and $\frac{\|\sum_{i=1,2,3} \boldsymbol{p}_i \times \boldsymbol{f}_i\|_2}{\frac{1}{3}\sum_{i=1,2,3}\|\boldsymbol{p}_i \times \boldsymbol{f}_i\|_2} \times 100\%$. If wrench closure is achieved, both ratios should be zero. All ratios are reported as *median, (25th percentile, 75th percentile)*. We computed these ratios for grasps planned with our method and CVAE-kinematics ablations. These ratios are not computed for CVAE-only because there are often fewer than 3 finger-object contacts. Our method achieved a median of $\leq 0.01\%$ on both ratios, showing that BO is effective in enforcing wrench closure. We note that while the CVAE+kinematics ablation does not explicitly consider external wrench, it still produced many grasps that achieve wrench closure by coincidence. This reflects CVAE's ability to produce *qualitatively* correct grasps.

To show that physical feasibility is a necessary condition for a successful grasp, we examined the CVAE+kinematic trials, which may not satisfy dynamic constraints, on "ramen" and "mustard bottle." We also executed grasp plans that are reported to be infeasible by BO. Table 3 summarizes the results. There is a strong correlation between hardware success and wrench closure. Moreover, all grasps reported to be infeasible by BO failed on hardware. This confirms that grasp refinement derived from rigorous physics-inspired metrics can significantly improve the final grasp plan quality.

## 5 Limitations and Conclusion

**Limitations.** Our method requires an observation of the full object point cloud. As our choice of physical constraint formulation does not include explicit robustness margins, estimation errors from the vision pipeline is a major source of grasp failures. This may be addressed by introducing point cloud completion (e.g., [43, 44, 45]) or grasp robustness metrics (e.g., [17]). Additionally, it is currently not possible to specify which type grasps to generate (e.g., "top grasp" or "side grasp") with our method. Finally, our method currently assumes that the object to grasp is placed on a flat surface and that there are no other objects in the scene.

**Conclusion.** This work presents a novel pipeline that combines learning-based grasp generation with bilevel optimization to produce diverse and physically feasible dexterous grasps. Our method achieved a grasp success rate of $86.7\%$ on 20 challenging real-world objects. Ablation studies demonstrated that an integrative approach combining learned models and rigorous physics-inspired metrics can achieve superior grasp output quality. Grasps initially generated by CVAE may not satisfy all physical constraints. However, by incorporating bilevel optimization for grasp refinement and rejection sampling, poor grasp predictions can be removed. This paradigm of combining learning and physics-inspired bilevel optimization may be applied to other robotic manipulation tasks.

**Acknowledgments**

We would like to thank Oussama Khatib, Jeannette Bohg, Dorsa Sadigh, Samuel Clarke, Elena Galbally Herrero, Wesley Guo, and Yanchao Yang for their assistance on setting up the hardware experiments. We would also like to thank Chen Wang and Jiaman Li for advice on training the CVAE model. Our research is supported by NSF-NRI-2024247, NSF-FRR-2153854, Stanford-HAI-203112, and the Facebook Fellowship.

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
