# OpenReview forum: "Learning Diverse and Physically Feasible Dexterous Grasps with Generative Model and Bilevel Optimization"
_robot-learning.org/CoRL/2022/Conference — CoRL 2022 Poster_

### Official Review · Reviewer_GKLm · 2022-07-29

**Originality:** Good
**Technical Quality:** Good
**Clarity Of Presentation:** Very Good
**Impact:** 3

**Recommendation:**

Weak Accept: I recommend accepting the paper, but will not argue for my recommendation if the majority of other reviewers have a different opinion.

**Summary:**

The paper presents a grasp planning pipeline comprised of an initial guess from the object geometry and a subsequent optimization to refine its physical feasibility.
The initial guess of promising grasps are generated from the prior of a learned conditional VAE over 6 YCB object point clouds.
The refinement optimization is based on force-closure criteria and specifically decomposed into two levels to exploit pyramidal friction cones for a QP structure at the lower-level.
The experiments include real robot grasps for seen, novel objects and ones with adversarial features. The pipeline demonstrates success rates between 55%-70% and is compared to hardcoded and CVAE only baselines.

**Issues:**

- relevance to robot learning
- technical novelty
- lacking adequate evidence and discussion about the role of the contributed planning method

**Quality Of The Limitations Section:**

Additional details required

**Reviewer Expertise:**

4: The reviewer is confident but not absolutely certain that the evaluation is correct

**Robotics Focus:**

Sufficient demonstration on hardware

**Strengths And Weaknesses:**

Strength:

The paper execution is clear. The technical idea is neat for handling non-affine equality constraints via a bilevel treatment and avoiding initial guess of finger-tip forces. Real robot experiment is carried out.

Weakness:

Significance and relevance to robot learning: the only learning part of the paper is training a CVAE as a generative model to sample roughly correct grasps.
The main technical contribution is closer to grasp planning and the paper content does not highlight the significance of CVAE and its necessity to other alternatives in the pipeline. The data and generalization setups look not very extensive.
Also, the idea of learning a generative model for synthesizing diverse grasps or potential actions is not novel, see some previous works using non-NN generative model [1][2].

Experimental results: it is hard to tell how successful the proposed pipeline is from the main chart Table 1 and 2. Interestingly, the proposed pipeline is even not performing as well as the hardcoded baseline for the category of seen objects.
I am always wondering to what extent qualitative results like in Figure 4 can be attributed to the exact planning results. As far as I know, the physical realization of a grasp tends to be disturbed from its nominal result due to the gap of finger contacts and inaccurate execution on the Allegro hand (lacking compensation for inertia, joint frictions, etc.).
A static grasp does not specify how the robot hand should approach the object and whether finger placement could be well synchronized as such the grasped object will not be tipped away from the intended pose.
The paper can be greatly improved if it could decisively show that it is the planner instead of some controllers doing much of the heavy lifting here. This could at least help to address why we are observing a mixed message from Table 1 and 2.

[1] Huang et al, Learning a real time grasping strategy, ICRA 2013

[2] Sun et al, Characterizing Continuous Manipulation Families for Dexterous Soft Robot Hands, Frontiers in Robotics and AI 2021


**Summary Of Recommendation:**

The paper contains some good technical ideas and the execution is generally smooth. However, the paper, at least in its current form, only shows a tenuous relevance to learning methods. The results are also not very conclusive and open to interpretation.

[Update on the version of Aug 28th] After seeing the authors' rebuttal, I am more leaning towards borderline so the scores are updated.

---

> ### Author Response · Authors · 2022-08-23
> **Response to comments from reviewer GKLm**
>
> > Significance and relevance to robot learning: the only learning part of the paper is training a CVAE as a generative model to sample roughly correct grasps. The main technical contribution is closer to grasp planning and the paper content does not highlight the significance of CVAE and its necessity to other alternatives in the pipeline. The data and generalization setups look not very extensive. Also, the idea of learning a generative model for synthesizing diverse grasps or potential actions is not novel, see some previous works using non-NN generative model [1][2].
>
> Please refer to section “learning component” on how CVAE is an integral part of the pipeline. We provide additional analysis below on why CVAE is irreplaceable by other methods.
>
> It is well established in the classical grasping literature (e.g. [a][b]) that the set of feasible grasps lie on a highly complex manifold defined implicitly by the physical constraints. Choosing the right region of the manifold, which is a combinatorially complex problem, is necessary for selecting favorable grasp configurations. In our approach, the region choice is provided by CVAE. Bilevel optimization ensures our produced grasp is indeed on the manifold once the initial region has been chosen.
>
> A deep generative approach such as CVAE is crucial for this region choice. The model must be able to account for the vastly different numbers of manifold regions for each object. As a simple illustration, the number of possible finger-face contact combinations for 3-finger object grasping a tetrahedron is at most $4^3= 125$, but for a box it is $6^3=216$. The representation power of classical generative models likely cannot capture this complexity.
>
> While we acknowledge that the idea of learning a generative model for grasp synthesis is not novel, an effective and theoretically rigorous implementation of the idea does not exist in the literature to the best of our knowledge. In both [1] and [2], the number of Gaussian components are fixed across objects. The robotic hands used for planning in [1] (9dof iCub and 4dof Barrett) have fewer degrees of freedom than ours, so the grasp planning problem is significantly easier. In fact,  [1] states “the success rate depends on the dimensions of the grasp space and the surface geometry of the target objects. Grasps in lower degrees of freedom (the Barrett hand) have higher success rates than those in higher degrees of freedom (the iCub hand)”. Moreover, neither [1] nor [2] contains any hardware experiments. The effectiveness and practicality of their proposed methods is thus unclear.
>
> [a] Rosales, Carlos, Josep M. Porta, and Lluıs Ros. "Global optimization of robotic grasps." Proceedings of robotics: science and systems VII 3 (2011).
> [b] Hang, Kaiyu, et al. "Combinatorial optimization for hierarchical contact-level grasping." 2014 IEEE International Conference on Robotics and Automation (ICRA). IEEE, 2014.
>
> > Experimental results: it is hard to tell how successful the proposed pipeline is from the main chart Table 1 and 2. Interestingly, the proposed pipeline is even not performing as well as the hardcoded baseline for the category of seen objects. I am always wondering to what extent qualitative results like in Figure 4 can be attributed to the exact planning results. As far as I know, the physical realization of a grasp tends to be disturbed from its nominal result due to the gap of finger contacts and inaccurate execution on the Allegro hand (lacking compensation for inertia, joint frictions, etc.). A static grasp does not specify how the robot hand should approach the object and whether finger placement could be well synchronized as such the grasped object will not be tipped away from the intended pose. The paper can be greatly improved if it could decisively show that it is the planner instead of some controllers doing much of the heavy lifting here. This could at least help to address why we are observing a mixed message from Table 1 and 2.
>
> Please refer to the “baselines” section and “connection between theoretical guarantees of physical feasibility and evaluation metric” for details.

---

> > ### Comment · Reviewer_GKLm · 2022-08-25
> > **More insights on CVAE as the core learning contribution**
> >
> > I would like to thank authors for the extensive responses and clarifications.
> >
> > First I probably should clarify that, the reason I brought up [1] is that I wanted to point out that using generative models to obtain diverse grasps exists even before the flourish of deep neural nets so I can imagine CVAE must have been used as an "upgrade" in more recent literature as in [7]. Unfortunately, we cannot gain much further insight about encoding feasible grasp configurations from this submission after [7] since they are using the exact same model for the same purpose. From the learning perspective, the paper could have been much more impactful if it include an analysis about the topology and geometry of multifingered grasps' solution space. It is well known that VAE in general cannot transform original data domain to a more well-behaved one such as Euclidean, and interpolating in the latent space will result in out-of-distribution samples [a]. So why CVAE can be regarded as a solid choice for encoding disconnected and complex grasp space when we have other adversarial, flow and diffusion-based models in hand? Without a theoretical motivation about the core learning contribution, the submission can only resort to empirical evidence which, unfortunately, is not significant given the current results and lacks a comparison to other generative models.
> >
> > Regarding the refinement optimization, anecdotally I agree with the authors' experience that, without informative initial guess, solving grasping as a single optimization is hard. Nonetheless, it is doable with a meticulous manipulation of constraints for an equivalent or relaxed form and I would think what proposed in this paper can also be classified as constraint reformulation. From a system integration perspective, however, the question is how crucial is it to rigidly fulfill these constraints for a successful physical grasp. The awkward fact is that the proposed pipeline even falls behind hard-coded grasp for seen objects. Honestly I kinda expect this given my personal experience on implementing grasps on physical systems like Allegro hands. But this also raises: if the object can be well lifted by a planner roughly placing finger-tips in a tripod pose and applying a squeeze control, how can we be sold that conforming to force-closure constraints without any slackness is a must-have feature?
> >
> > Still, I want to emphasize that I like the technical treatment using nested QP and I think current progress on differentiable optimization can broaden our toolkits much for problem formulation. The benefits of proposed method may be better demonstrated on cooperative manipulation, where the physically feasible poses can be more accurately realized and more significant given heavier loads.
> >
> > a. Shao et al, The Riemannian Geometry of Deep Generative Models, CVPR 2018

---

### Official Review · Reviewer_Fnic · 2022-07-30

**Originality:** Good
**Technical Quality:** Very Good
**Clarity Of Presentation:** Very Good
**Impact:** 3

**Recommendation:**

Weak Accept: I recommend accepting the paper, but will not argue for my recommendation if the majority of other reviewers have a different opinion.

**Summary:**

This paper presents a framework for dexterous grasp of a single object in uncluttered scenes. Specifically, the authors first trained a conditional variational auto-encoder (CVAE) to generate an initial set of contacts points given the observed object point cloud and a feasible grasp from the dataset. This will reduce the combinatorially complex contact choices when solving by an analytical approach. However, it cannot guarantee the contact points are physically feasible. Therefore, the authors proposed to use bilevel optimization to map the contact points predicted by CVAE onto the manifold of kinematically and dynamically feasible grasps. The proposed framework is evaluated in real world with a Franka Emika Panda arm and an Allegro hand. Compared with the other two baseline methods, i.e., hardcoded and CVAE-only framework, the proposed method achieved better or comparable results on seen and adversarial objects.

**Issues:**

See above.


Additional:
- What is the reason to train the model on only 5 objects?
- What is the computation time of the overall system and each component?


**Quality Of The Limitations Section:**

Limitations are addressed clearly

**Reviewer Expertise:**

4: The reviewer is confident but not absolutely certain that the evaluation is correct

**Robotics Focus:**

Sufficient demonstration on hardware

**Strengths And Weaknesses:**

Strengths
- This paper presented a well designed framework for dexterous grasping using raw point cloud as input, and conducted real-world robots experiments which demonstrated the proposed method can be applied to both seen and unseen objects.
- The paper is well organized and the writing is clear and easy to understand.
- I also appreciate the details and the videos in the supplementary materials.

Weaknesses
- Though I like the practicality of the proposed system, the novelty of the paper might be limited. Predicting contacts or contact points is not a novel idea and has been explored in many papers, such as [14, 17, 18], and [a, b]. Using CVAE to predict grasp is also used in [7], but I think the idea of generating grasps conditioned on a grasp from the dataset is interesting and can constrain the predicted grasps. The authors could add more discussion and analysis on this.
- Bilevel optimization for dexterous grasp might be novel but using analytical approach to refine/solve the prediction/initialization from a learning-based approach is not novel.
- Though the proposed framework seems well designed and engineered, it did not outperform the hardcoded baseline: it is hard for me to tell which one is better. Therefore it is difficult to verify the effectiveness of the proposed method.
- I am also curious why the results of hardcoded and CVAE-only baselines are missing in Table 2.

References
- [a] Brahmbhatt, Samarth, et al. "Contactdb: Analyzing and predicting grasp contact via thermal imaging." Proceedings of the IEEE/CVF conference on computer vision and pattern recognition. 2019.
- [b] Brahmbhatt, Samarth, et al. "Contactgrasp: Functional multi-finger grasp synthesis from contact." 2019 IEEE/RSJ International Conference on Intelligent Robots and Systems (IROS). IEEE, 2019.

**Summary Of Recommendation:**

**Final comment: thanks to the authors for the additional experiments and discussions to address my concerns and other reviewers' questions. I will keep my rating unchanged.**

**Minor: Table 1-3. It'll be more "fair" if the hard-coded one also has 6 trials.**

----
(initial comment)
The paper is well written and the real-world experiments look good. But the technical novelty is a bit limited. My rating is more on the boarder line.

---

> ### Author Response · Authors · 2022-08-23
> **Response to comments from reviewer Fnic**
>
> > Predicting contacts or contact points is not a novel idea and has been explored in many papers, such as [14, 17, 18], and [a, b]. Using CVAE to predict grasp is also used in [7], but I think the idea of generating grasps conditioned on a grasp from the dataset is interesting and can constrain the predicted grasps. The authors could add more discussion and analysis on this.
>
> > Bilevel optimization for dexterous grasp might be novel but using analytical approach to refine/solve the prediction/initialization from a learning-based approach is not novel.
>
> Please refer to the “novelty of our work” section for more discussion on these two concerns. We will also update our manuscript with the content there.
>
> > Though the proposed framework seems well designed and engineered, it did not outperform the hardcoded baseline: it is hard for me to tell which one is better. Therefore it is difficult to verify the effectiveness of the proposed method.
>
> Please refer to the “baselines” section for explanation on our baseline choice and its implications.
>
> > I am also curious why the results of hardcoded and CVAE-only baselines are missing in Table 2.
>
> We will add these results.
>
> > What is the reason to train the model on only 5 objects?
>
> We could train the model on more objects, but we found this to be sufficient.
>
> > What is the computation time of the overall system and each component?
>
> We will add the statistics shortly. Please stay tuned.

---

### Official Review · Reviewer_XLAr · 2022-07-30

**Originality:** Fair
**Technical Quality:** Good
**Clarity Of Presentation:** Good
**Impact:** 2

**Recommendation:**

Weak Reject: I recommend rejecting the paper, but will not argue for my recommendation if the majority of other reviewers have a different opinion.

**Summary:**

Proposed using bilevel optimization to hand dexterous grasping problems with three fully actuated fingers. Provided derivations to the grasp planning pipeline with further verifications in experimental results. Considered both kinematic and dynamic constraints in performing the grasping task. Unclear how learning is integrated into the proposed method, at least the integration with CVAE is relatively marginal, contributing to the proposed grasping method. But there is a lack of sufficient details to explain the bilevel optimization, which is more prone to optimization.

**Issues:**

- Explain clearly how learning contributes to the proposed method for robotic manipulation.
- Explain the bilevel optimization method with further details. A brief paragraph is not enough.
- How much time would it cost to generate a grasp planning solution? What is the success rate for generating a grasp planning?
- The method is rather complex to reproduce without code. Please provide and justify how would others reproduce the results.
- Please add (significantly) more grasp trails to enlarge the sample size of the reported results.
- While the authors used dynamics in the derivation while performing the grasping, the authors explained, “A fixed, hand-designed squeezing motion is performed after q* has been achieved to provide contact forces.” If the squeezing motion is achieved through a fixed, hand-designed command, then what’s the point of incorporating dynamics in grasp planning? In other words, the proposed method does not actually provide actual use during grasping.
- It is strange that the “massage ball” shows poor results, and no tactile sensors are used on the fingertip. So, how does the force used in performing the calculation or during the execution? How does the proposed compensate for material softness based on the presented method? How to justify that if not using the proposed method, massage balls like these are usually quite simple to be picked up by a three-finger gripper.


**Quality Of The Limitations Section:**

Additional details required

**Reviewer Expertise:**

4: The reviewer is confident but not absolutely certain that the evaluation is correct

**Robotics Focus:**

Sufficient demonstration on hardware

**Strengths And Weaknesses:**

Strength:
The use of bilevel optimization in dexterous grasping is interesting and is backed by analytical reasoning from both dynamics and kinematics constraints. The application of the resultant method in robotic grasping is also a plus.

Weakness:
- It is unclear how learning is integrated based on the current content, more like optimization based on the current content.
- The proposed method, bilevel optimization, lacks sufficient details to explain what it is. Only one short paragraph is not enough, especially when this is the core method behind this paper. The derivation part helps but not sure what it means by bilevel or how it contributes.
- The analytical part is excellent, but more experiments are needed. The method is already complex enough based on the current content, making it suspicious to see how much time would it cost to generate a grasp (any advantage over existing analytical or learning-based methods?) At least, the grasp planning time is not reported in the current paper.
- The authors mentioned many advantages during deriving the method but later explained that getting a working solution for grasping is not an easy task. This is also confusing but not surprising as the method is simplified and more complex. Why this method is not yet used in literature? Any superior advantage?
- The experiment results are suspicious, many 66.7% (2/3), 83.33% (5/6), 50% (1/2), 33.3% (1/3), then another one with 94.44%. In short, the sample size is too small, and the authors should have conducted more grasping trials to further justify if this result is consistent or just luck.
- It seems the CVAE is the only learning-related part. Not sufficient based on the current content to be learning focused on robotic manipulation or the scope of the conference.


**Summary Of Recommendation:**

The main problem is probably the complexity of the proposed method itself is not thoroughly learning-related, lacks comparison with other dexterous grasping methods, without sufficient experimental trails to support the claimed results (even the authors admitted that the grasp planning success rate is challenging), and not sure how would such method be used to strength contribution of learning in robotics.

---

> ### Author Response · Authors · 2022-08-23
> **Response to comments from reviewer XLAr (1/2)**
>
> > It is unclear how learning is integrated based on the current content, more like optimization based on the current content.
>
> We emphasize that learning is an inseparable component in our method, and our primary contribution is the systematic integration of learning and optimization. As with most nonconvex optimization problems, the bilevel optimization needs a reasonable initial guess to converge. Moreover, the initial guess must be able to capture the diversity of possible grasps. It is very challenging for a gradient-based optimizer to explore across different grasp modalities . Thus, choosing a generative model is necessary and is key to our method’s ability to produce diverse grasp configurations for many objects.
>
> Please also refer to the section “learning component” in the general response.
>
> > The proposed method, bilevel optimization, lacks sufficient details to explain what it is. Only one short paragraph is not enough, especially when this is the core method behind this paper. The derivation part helps but not sure what it means by bilevel or how it contributes.
>
> We agree that bilevel optimization is a critical contribution of our method and we will add more details to the manuscript. For a review on the established bilevel optimization mathematics, we recommend the following references:
> - Landry, Benoit, et al. "Bilevel optimization for planning through contact: A semidirect method." The International Symposium of Robotics Research. Springer, Cham, 2019.
> - Sinha, Ankur, Pekka Malo, and Kalyanmoy Deb. "A review on bilevel optimization: from classical to evolutionary approaches and applications." IEEE Transactions on Evolutionary Computation 22.2 (2017): 276-295.
>
> > The authors mentioned many advantages during deriving the method but later explained that getting a working solution for grasping is not an easy task. This is also confusing but not surprising as the method is simplified and more complex. Why this method is not yet used in literature? Any superior advantage?
>
> We speculate the main reason why bilevel optimization has not been widely applied is due to the limitation of computational tools. The library that implements differentiation through quadratic programs, which is the core of our bilevel optimization implementation, was released in 2017 [1]. Additionally, we leveraged automatic differentiation extensively through the library Drake [2] to avoid needing to derive complex rigid body equations and their gradients for the purpose of optimization. Since differentiable optimization is not readily available in standard physics simulators, we had to implement the automatic differentiation interface between OptNet and Drake to obtain the desired bilevel optimization formulation.
>
> [1] Amos, Brandon, and J. Zico Kolter. "Optnet: Differentiable optimization as a layer in neural networks." International Conference on Machine Learning. PMLR, 2017.
> [2] Tedrake, Russ, The Drake Development Team. ‘Drake: Model-based design and verification for robotics’. 2019. Web.
>
> > How much time would it cost to generate a grasp planning solution? What is the success rate for generating a grasp planning?
>
> We will add these statistics shortly. Please stay tuned.
>
> > The method is rather complex to reproduce without code. Please provide and justify how would others reproduce the results.
>
> We plan to release the full source code for training and execution, as well as the relevant system details. The only custom hardware in our system is a 3D-printed mount for attaching the Allegro hand to the Franka robot, which could easily be replaced by off-the-shelf hardware. All other components are commercially available and our code can be applied directly.
>
> > Please add (significantly) more grasp trails to enlarge the sample size of the reported results.
>
> We are running the experiments and will update our submission with the new results later in the rebuttal period.
>
> ------
> (continued on the next comment due to character limit)

---

> ### Author Response · Authors · 2022-08-23
> **Response to comments from reviewer XLAr (2/2)**
>
> > While the authors used dynamics in the derivation while performing the grasping, the authors explained, “A fixed, hand-designed squeezing motion is performed after q* has been achieved to provide contact forces.” If the squeezing motion is achieved through a fixed, hand-designed command, then what’s the point of incorporating dynamics in grasp planning? In other words, the proposed method does not actually provide actual use during grasping.
>
> The grasp planner ultimately produces the grasp configuration (joint angle) and a set of possible contact forces (“internal” decision variables abstracted away in the lower level of the bilevel optimization, denoted as $f_i$). However, once a finger placement is chosen, the valid grasp forces are not unique, and the robot does not need to execute the specific $f_i$ as long as the executed one is still valid. The main purpose of our method is to avoid grasp configurations where *no* possible contact force exists, i.e. the grasp is infeasible.
>
> That said, we have changed our hand-designed squeezing motion to a Jacobian-based implementation that squeezes in the direction and magnitude ratios of the planned contact forces.
>
> > It is strange that the “massage ball” shows poor results, and no tactile sensors are used on the fingertip. So, how does the force used in performing the calculation or during the execution? How does the proposed compensate for material softness based on the presented method? How to justify that if not using the proposed method, massage balls like these are usually quite simple to be picked up by a three-finger gripper.
>
> The massage ball is a rigid object; all of the test objects are rigid. We will clarify this in the revised manuscript. We suspect the failure may stem from poor visual estimation.

---

### Official Review · Reviewer_SirZ · 2022-07-30

**Originality:** Good
**Technical Quality:** Fair
**Clarity Of Presentation:** Very Good
**Impact:** 2

**Recommendation:**

Weak Reject: I recommend rejecting the paper, but will not argue for my recommendation if the majority of other reviewers have a different opinion.

**Summary:**

This paper adopts a CVAE network to predict grasp pose for dexterous hand and optimizes it with a bilevel optimization to force grasp pose to be more physically feasible.

**Issues:**

1.	Illustrate why not consider the work that considers either kinematic[1][2] or dynamic constraints[3] as the baseline, or provide the improvement of the method compared with these methods
2.	Provide more metrics to show “physically feasible” either in simulation or real. Such as measuring the penetration before and after optimization in simulation.
3.	Ablation study should include the experiment to show how these two-level optimizations contribute to the final result, e.g., which one might be more important?


**Quality Of The Limitations Section:**

Limitations are addressed clearly

**Reviewer Expertise:**

5: The reviewer is absolutely certain that the evaluation is correct and very familiar with the relevant literature

**Robotics Focus:**

Sufficient demonstration on hardware

**Strengths And Weaknesses:**

Strengths:
1.	This paper considers both kinematic and dynamic constraints for a more physically feasible grasp pose which seems novel to me, as far as I know, this is the first to consider both two levels.
2.	This paper conducts rich real-world experiments.
Weaknesses:
1.	Baseline seems missing: Since the main contribution of this paper is bilevel optimization, the author should compare with the baseline that only uses one level to optimize. Such as [1][2], consider the collision, [3] consider force closure. However, the author seems not to compare the method with these baselines.
[1] Wei, Wei, et al. "DVGG: Deep variational grasp generation for dextrous manipulation." IEEE Robotics and Automation Letters 7.2 (2022): 1659-1666.
[2] Liu, Min, et al. "Generating grasp poses for a high-dof gripper using neural networks." 2019 IEEE/RSJ International Conference on Intelligent Robots and Systems (IROS). IEEE, 2019.
[3] Liu, Tengyu, et al. "Synthesizing Diverse and Physically Stable Grasps With Arbitrary Hand Structures Using Differentiable Force Closure Estimator." IEEE Robotics and Automation Letters 7.1 (2021): 470-477.
2.	Evaluation metric: although the author mainly focuses on physically feasible grasp, the metric in this paper is success rate without metric about how to measure the grasp is more physically feasible, if only compare success rate, other methods which may not consider physically feasible may also achieve success rate. At least in simulation, I think the author should use other metrics, such as penetration for depth or volume used in [1].
3.	Since the main difference is bi-level optimization, ablation should show how these two levels contribute to the final result.


**Summary Of Recommendation:**

The author mainly proposes a bilevel optimization to generate a more physically feasible grasp pose which is novel. However, generating a grasp pose with CVAE is not novel, since other work[1] has already tried it.  Although this paper focuses on physically feasible, the metric for evaluation is only success rate, which can hardly show the “physically feasible”. There are also other methods that either consider kinematic or dynamic constraints which should be considered as the baseline, however, the author only conducts an ablation study that only compare w and w/o optimization, not showing which level is more important. Thus, I recommend this paper as a weak reject.

---

> ### Author Response · Authors · 2022-08-23
> **Response to comments from reviewer SirZ**
>
> > Baseline seems missing: Since the main contribution of this paper is bilevel optimization, the author should compare with the baseline that only uses one level to optimize.
> > Illustrate why not consider the work that considers either kinematic[1][2] or dynamic constraints[3] as the baseline, or provide the improvement of the method compared with these methods
>
> Our method is fundamentally different from the single-level optimization papers as our optimization has no constraint relaxation. The listed papers (among many single-level optimization works) use “soft” constraints, where a single objective value defined as the sum of loss terms is minimized. However, the minimizing objective value does not necessarily represent constraint satisfaction. For instance, consider the objective value of $||$ finger-object signed distance $||$ + regularization. It is possible that the regularization term is dominating enough that a small distance error is allowed.
>
> We would like to reiterate that formulating a single-level optimization with “hard” constraints is extremely difficult to solve. Anecdotally, our single-level implementation of the grasp optimization with kinematic and dynamic constraints did not solve at all. The problem is only tractable with the bilevel formulation.
>
> Please also refer to section “novelty of our work” in the general response for more details.
>
> > Provide more metrics to show “physically feasible” either in simulation or real. Such as measuring the penetration before and after optimization in simulation.
>
> Please refer to “Connection between theoretical guarantees of physical feasibility and evaluation metric" in the main response. In addition, we will add simulation/planning metrics such as computed contact force and finger penetration depth to our results.
>
> > Ablation study should include the experiment to show how these two-level optimizations contribute to the final result, e.g., which one might be more important?
>
> The bilevel optimization is a single unit of formulation. It is possible to remove the lower level entirely, leaving only the kinematic feasibility problem as an “ablation.” This result is included in the original manuscript under the “CVAE only” group of experiments, which has much lower success rates compared to the full bilevel optimization. Singling out the lower level (force closure) subproblem is not meaningful as many trivial solutions that are clearly not feasible exist.

---

### Author Response · Authors · 2022-08-28
**New experiment results and revised manuscript (1/2)**

To address the concerns with our evaluation, we significantly scaled up our experiments and replaced the ad hoc squeezing controller with a Jacobian-based controller that aims to faithfully execute the planned contact forces. Excluding ablation and baseline experiments, **we completed 120 grasps across 20 objects for our method and achieved an overall grasp success rate of 86.7%. We now significantly outperforms the baseline, which achieved an overall success rate of 53.3%.**

We list our new experiments by the concerns they address below. We also included new data tables excerpted from the manuscript. For more details, please refer to our revised manuscript. New sections, tables, and figures added for rebuttal are shown in blue.

# Dataset Size
- We **increased the number of repeats from 3 to 6 for all experiments involving our method** and the ablations for improved statistical significance.
- Additionally, we **added 6 novel objects** (hair spray, pear, alcohol bottle, condiments bottle,  sandwich box (side), lego stack) to reach a total of **20 test objects**.
- The additional novel objects have significantly different geometries from the training objects and require diverse classes of grasps. These results once again showcase the generalizability of our method to different real world objects.

# Baseline
- We expanded our hardcoded baseline experiments to cover all objects tested.
- The baseline method achieved 53.3% grasp success rate across all objects. However, **among the novel objects, the success rate is only 38.5%. This is significantly worse than our method’s success rates of 86.7% overall and 82.1% on novel objects**.

# Ablations
- To provide a better comparison, we updated our ablation design to include the following
    - *CVAE-only*: only solve a collision-free IK to match the allegro fingertips to the CVAE predicted finger placements. We believe this is
closest to a “pure learning” approach with CVAE.
   - *CVAE-kinematics*: solve the “upper” level of the bilevel optimization, which included collision avoidance and finger-object contact constraints. Consideration for force closure (the “lower” level of the bilevel optimization) is ablated. This is the original ablation in our paper.
- The ablations were tested on 3 objects seen during CVAE training. **CVAE-only achieved 38.9% successes, and CVAE-kinematics achieved 88.9% successes. The full method we proposed achieved 94.4%**
- We reported grasping metrics for the ablations and found the results to match the ablated components. See “Grasp metrics for physical feasibility” for more details

# Grasp metrics for physical feasibility
- We computed finger penetration depth to the fitted object meshes
    - The fingertip-object signed distances achieved in grasps produced by our method is as prescribed (subjected to numerical tolerance). **The maximum constraint violation is 0.082cm**, much more precise than other possible error sources in the pipeline, e.g. point cloud visual estimation.
    - As expected, CVAE + kinematics ablation experiments, which also considers kinematic constraints, achieved comparable fingertip-object signed distances to the full approach.
    - The range of finger-object distances resulting from the CVAE only ablation is much larger (max 2.032cm constraint violation), indicating that while CVAE alone can produce qualitatively correct grasp, it cannot enforce the contact constraints.
    - The merits of enforcing contact constraints is backed by the 38.9% grasp success rate in CVAE only experiments vs. 88.9% (CVAE+kinematics) and 94.4% (ours) on the same set of objects.

- We added force and torque closure metrics that computes the ratio between the external force and torque applied on the object versus the average magnitude of the applied contact force and torque. For CVAE+kinematics, we compute the “best contact force that could be applied given the finger placement” after the grasp has been planned.
    - Our method achieved an **overall median of <0.01% external force ratio and 0.01% external torque ratio**, validating the merits of our dynamic constraint formulation (subjected to numerical tolerance).
    - The force and torque closure metrics shows that including dynamic constraints with our bilevel formulation is very effective in eliminating the external wrench in the planned grasp.
    - While the CVAE+kinematics ablation does not consider the external force and torques, it may achieve force closure by coincidence. This again represents CVAE’s ability to produce qualitatively correct grasps. We discuss this in detail in “Connection between theoretical guarantees of physical feasibility and hardware success”
    - Force and torque closure metrics are not computed for CVAE only because the contact normals are not defined if there is no contact.

---

> ### Author Response · Authors · 2022-08-28
> **New experiment results and revised manuscript (2/2)**
>
> # Connection between theoretical guarantees of physical feasibility and hardware success
> - We executed CVAE predictions rejected by bilevel optimization on “ramen” and “mustard bottle”. **All rejected grasp predictions failed on hardware**. The median of the maximum external force/torque % on the grasp is 65.90%
> - We attempted CVAE+kinematic grasps on “ramen” and “mustard bottle”. We observed a **strong correlation between grasp success and force/torque closure of the planned grasps**. The median of the maximum external force/torque % on the grasps is 0.0% for grasps that succeeded on hardware and 50.71% for grasps that failed.
> - In other words, all grasps that succeeded achieved force and torque closure, all grasps that failed lack either force or torque closure
> - This showcases how rejection sampling derived from rigorous physics-inspired metrics can significantly improve the final model output quality.
>
> # Faithfulness of hardware execution
> - We provide **visual comparison of the grasp executed on hardware versus the planned grasp** to showcase our hardware execution’s faithfulness to the planned grasps.
> - We **replaced the ad hoc squeezing motion with a Jacobian-based squeezing controller motivated by impedance control** to accurately execute the planned contact forces. All experiments affected by this change were reperformed accordingly.
>
> # Timing statistics
> - Timing statistics have been reported for each component of our method. The median time to produce a grasp with our method is 14.4 seconds.
> - To generate a grasp, our method samples 2 CVAE latent variables, solves 2 IKs, and solves 2 bilevel optimizations as suggested by the median.

---

### Meta-Review · Area_Chair_LTAv · 2022-08-15

**Recommendation:** Accept (Poster)
**Confidence:** 4

**Metareview:**

This paper proposes a new grasp learning strategy that takes into account both grasp pose prediction and grasp kinematic and dynamic constraints, to achieve better stable grasp behaviors. The idea is nicely carried out, the use of  bi-level optimization in dexterous grasping is interesting and relevant for robot learning. Experiment on real robots is very positive. The authors have made great response to clarify doubts and open questions raised by the reviewers. For completeness, the authors should elaborate these justifications in the final version of the paper.



**Best Paper Nomination:**

No

---

> ### Author Response · Authors · 2022-08-24
> **Overall response: addressing comments and upcoming improvements (1/2)**
>
> ------
> We discovered a mistake in setting the comment permissions when we first posted our comments. As of now (8/24) we have posted the following
> 1. Overall response
> 2. Reviewer specific response for each of the 4 reviewers (SirZ, XLAr, Fnic, GKLm)
>
> Please let us know if there is anything missing. We apologize for the delay in releasing our response.
>
> ------
>
> We thank the reviewers for the thoughtful comments and feedback. Here we provide a general address to the common concerns. Our answers to specific questions raised by each reviewer are posted in detail in the reviewer-specific threads.
>
> # Experiment data
> We are performing more experiments and repetitions to improve statistical significance. We will update our submission with the new results later in the rebuttal period.
>
> # Novelty of our work
> We presented an integrative approach that combines learning and model-based methods for dexterous grasping. While some of the individual components may be similar to literature, our innovation lies in the unique combination and the design decisions that make our method theoretically justified and effective in practice.
>
> ### Compared to other works in learned grasp generation
> While the idea of using generative methods to produce grasps is not new, most of the literature do not contain hardware experiments (e.g. [1][2][3][4][5]) or focus on simpler grippers with fewer degrees of freedom ([6]: parallel jaw gripper,  [4]: 10dof). Other papers explicitly classify the types of grasps (e.g. [7] splits grasps into “overhead” and “side”) instead of performing unrestricted grasp generation.
>
> The most relevant prior work in the grasp generation domain is likely DVGG[1] (published 2022/04). The grasping pipeline of DVGG is as follows: 1. perform point cloud completion on the target object 2. generate grasps with CVAE 3. iterative grasp refinement with a learned refinement module. However, DVGG requires a much larger training dataset (300 objects, ~1.5M annotated grasps), whereas our network was trained on simulated grasps of merely 6 YCB objects. Moreover, the grasp refinement module in DVGG is based on yet another neural network and does not provide theoretical guarantees. Meanwhile, our bilevel optimization grasp refinement is grounded in physical laws and provides theoretical certifications for the grasp predictions.
>
> ### Compared with other methods involving grasp refinement and optimization
> Generally speaking, most works in literature take the “relaxed” approach, where instead of enforcing each constraint directly, the violation of each constraint is cast as a scalar loss and summed together. This transforms the grasping optimization into an unconstrained optimization, which is significantly easier to solve than a constrained optimization. However, under such relaxation, there is no guarantee that the outcome of grasp optimization will satisfy the constraints that motivated the loss design. It is possible for the optimization to return a result with violated constraint despite the optimization being “feasible”. Moreover, the formulation of grasp optimization problems are often ad hoc. Some are motivated by physics modeling (e.g. [5],[9],[1]), while others train neural networks to predict grasp quality (e.g. [8]). Nevertheless, whether these formulations represent a good metric for selecting grasps remains unclear.
>
> ### Merits of our approach
> Our work uses CVAE to generate diverse initial grasp guesses on a fully-actuated dexterous hand without any need for manually classifying grasp categories. Our bilevel optimization formulation respects constraints obtained directly from physical laws and provides a concrete metric for performing rejection sampling on the CVAE samples. Not only does bilevel optimization refine specific CVAE grasps, an infeasible optimization also indicates that a new CVAE grasp sample should be generated. Thus, high-quality grasps can still be produced without requiring unrealistically precise model predictions.
>
> # Learning component
> Our primary contribution is an integrated approach that systematically combines learning and model-based methods. Solving the grasping problem exactly with all physical constraints as a single optimization is extremely difficult. While our bilevel formulation abstract away a significant part of the computation, as with most nonconvex optimization problems, the bilevel optimization needs a reasonable initial guess to converge. Hence CVAE is an inseparable part of our method as it provides the crucial initial guess. On the other hand, without the bilevel optimization, the CVAE prediction is not sufficient for performing successful grasps, as shown in our “CVAE only” ablation.
>
> ------
> (continued on the next comment due to character limit)

---

> ### Author Response · Authors · 2022-08-24
> **Overall response: addressing comments and upcoming improvements (2/2)**
>
> # Connection between theoretical guarantees of physical feasibility and evaluation metric
> We chose to evaluate our method using the success rate of the grasping task performed by a real dexterous robot, which should be the ultimate test for a grasp planner. However, we acknowledge that the guarantee of physical feasibility provided by our method is only one of the factors that contribute to the grasp success rate on hardware. Despite that, we have shown that physical feasibility is a critical factor to achieving a high grasp success rate, as evidenced by the "CVAE only" experiment. The “CVAE only” experiments serve as negative control with no consideration for physical feasibility guarantees. To further demonstrate that the success of our method is a result of the theoretical guarantees, we will add additional figures to show that our hardware grasp execution is true to the planned grasp.
>
> # Baselines
> A major advantage of our method is the ability to generate diverse grasps for many objects. Meanwhile, our baseline, which is a fixed top-down grasp tuned for a particular object, is primarily designed to illustrate the limitations of hand-engineered or pre-computation based methods. While it is possible to tune a grasp very well, such a grasp does not adapt to other scenarios, such as when side grasp or different finger placement is necessary. Indeed, our baseline is extremely successful in the lab setting for particular objects. However, our long-term goal is to develop a method capable of grasping in real-world scenarios where diverse grasps are necessary. For instance, approaching an object from the top might not be feasible due to obstacles. We acknowledge that our current experiment design does not showcase such a scenario as it requires designing collision-avoiding approach trajectories, which is not within the scope of our contribution.
>
> # References
> [1] Liu, Tengyu, et al. "Synthesizing Diverse and Physically Stable Grasps With Arbitrary Hand Structures Using Differentiable Force Closure Estimator." IEEE Robotics and Automation Letters 7.1 (2021): 470-477.
>
> [2] Brahmbhatt, Samarth, et al. "Contactdb: Analyzing and predicting grasp contact via thermal imaging." Proceedings of the IEEE/CVF conference on computer vision and pattern recognition. 2019.
>
> [3] Brahmbhatt, Samarth, et al. "Contactgrasp: Functional multi-finger grasp synthesis from contact." 2019 IEEE/RSJ International Conference on Intelligent Robots and Systems (IROS). IEEE, 2019.
>
> [4] Sun, Jiatian, Jonathan P. King, and Nancy S. Pollard. "Characterizing Continuous Manipulation Families for Dexterous Soft Robot Hands." Frontiers in Robotics and AI 8 (2021): 645290.
>
> [5] Grady, Patrick, et al. "Contactopt: Optimizing contact to improve grasps." Proceedings of the IEEE/CVF Conference on Computer Vision and Pattern Recognition. 2021.
>
> [6] Mousavian, Arsalan, Clemens Eppner, and Dieter Fox. "6-dof graspnet: Variational grasp generation for object manipulation." Proceedings of the IEEE/CVF International Conference on Computer Vision. 2019.
>
> [7] Lu, Qingkai, et al. "Multifingered grasp planning via inference in deep neural networks: Outperforming sampling by learning differentiable models." IEEE Robotics & Automation Magazine 27.2 (2020): 55-65.
>
> [8] Wei, Wei, et al. "DVGG: Deep variational grasp generation for dextrous manipulation." IEEE Robotics and Automation Letters 7.2 (2022): 1659-1666.
>
> [9] Liu, Min, et al. "Generating grasp poses for a high-dof gripper using neural networks." 2019 IEEE/RSJ International Conference on Intelligent Robots and Systems (IROS). IEEE, 2019.